# The tumor microenvironment of metastatic osteosarcoma in the human and canine lung
L. E. McGee [1] ✉, J. S. Pereira[2], T. A. McEachron[2], C. Mazcko[1], A. K. LeBlanc [1] & J. A. Beck[1]

Osteosarcoma is a rare but aggressive bone tumor that develops spontaneously in human and canine patients and most commonly metastasizes to the lung. The presence of lung metastases significantly decreases the survival rate of patients, with minimal benefit seen with available treatments. Canine osteosarcoma is clinically and molecularly similar to human osteosarcoma and develops approximately ten times more frequently than human osteosarcoma making dogs a promising natural model to study disease progression. The development of new therapies for pulmonary metastases requires an understanding of the interplay between tissue resident cells as well as recruited cell types and how those interactions impact seeding and progression within the new metastatic site. This review explores the tumor microenvironment surrounding pulmonary metastases and how current knowledge in canine and human patients can inform better treatments and outcomes for both populations.

Osteosarcoma is the most common form of bone cancer in children, young adults, and in dogs. Although survival for human patients with localized disease is approximately 70%, it drops to 20–30% for those that develop metastatic disease[1–3]. Both human and canine osteosarcomas have a proclivity for metastatic progression to the lung, which complicates treatment and decreases survival[4–6]. In human patients, survival rates have plateaued over the last 40 years[1–3,7]. Front-line therapeutic management for patients has also largely remained unchanged during this time frame, consisting of surgical resection for local tumor control and adjunctive combination of chemotherapeutics including doxorubicin and platinum agents[2]. Management of osteosarcoma in canine patients faces challenges like those of its human equivalent. Dogs receive comparable treatments including surgical resection with adjuvant chemotherapy. However, while human patients receive chemotherapy prior to amputation, canine patients undergo amputation before starting any chemotherapeutic treatments. It is likely that 80–90% of dogs present with micro-metastases in the lung at diagnosis, as dogs undergoing amputation alone have an 11.5% and 2% 1- and 2- year survival rate, respectively[8]. While this figure highlights the ability of the canine patient model to study metastatic progression, it should be noted that compared to disease-free interval, survival endpoints are impacted by an owner's decision to pursue euthanasia following disease progression[6,9,10].

Improving metastatic osteosarcoma outcomes is dependent on identifying and utilizing appropriate models for this facet of the disease. The osteosarcoma-bearing pet dog is an attractive model due to its 10-fold higher frequency and litany of molecular and biologic similarities including a predilection for pulmonary metastasis[6,11,12]. Further, companion dogs have an educated immune system with comparable cell types and functions and develop osteosarcoma in the same environment they share with their human companions[4,6,13]. Canine osteosarcoma patients present an ideal proving ground for new therapeutic strategies for humans, particularly in the realm of immunotherapy[14,15]. Dogs also offer an opportunity to study the presence of serum biomarkers that could indicate tumor progression or metastasis, including the presence of circulating tumor cells[16]. Ready access to tissue samples procured through veterinary clinical procedures and post-mortem examination, coupled with active comparative oncology trial networks, enable a range of studies that will expand our collective knowledge of metastatic progression and help to credential the dog as a patient model of human osteosarcoma.

With these similarities, studying osteosarcoma development and response to treatment in companion dogs provides an excellent opportunity to model the disease and improve outcomes for both human and canine patients. Treating lung metastases requires a thorough understanding of the functional interplay of the diverse cell types within the ever-evolving tumor microenvironment. Further study is needed to be able to adequately understand the role of the microenvironment in osteosarcoma lung metastasis to identify better therapies. Companion dogs are exposed to the same environment as their human companions, including environmental toxins or other hazardous exposures. Dogs also have an intact, similarly

¹Comparative Oncology Program, Center for Cancer Research, National Cancer Institute, National Institutes of Health, Bethesda, MD, USA. ²Pediatric Oncology Branch, Center for Cancer Research, National Cancer Institute, National Institutes of Health, Bethesda, MD, USA. ✉e-mail: mcgee.lauren.e@gmail.com

educated immune system, including tumors that contain a spectrum of immune infiltrates that influence treatment success and survival[14,17]. These biologic similarities provide a foundation for future studies that can benefit both human and canine patients.

## Resident lung cells and structures

Metastasis to a secondary organ is dependent on the development of a pre-metastatic niche that is cellularly and molecularly reprogrammed to support the colonization and outgrowth of the tumor cells prior to their arrival[18,19]. The lung is the most common site of metastatic progression in people and in dogs with osteosarcoma[20,21]. The tumor microenvironment of pulmonary metastases is comprised of multiple cell types and components including pneumocytes, fibroblasts, and endothelial cells (Fig. 1). These elements are altered by the initiation of the pre-metastatic niche and subsequent development of metastases, and in turn can impact the ability of the host immune system to either eliminate the metastases or promote immune evasion. This section focuses on the impact of metastases on the resident lung cells and structures, how the vascularization of the primary tumor can impact the pulmonary metastases, and how these findings can inform future research and be used to develop biomarkers in the canine that will translate back to human patients.

Pneumocytes, one of the resident lung cells lining alveolar spaces, have been implicated to play a role in both the premetastatic niche and in the progression of metastatic lesions within the lung. These cells are divided into two types: type I and type II[22,23]. Type I pneumocytes (TIPs) are the most abundant, cover much of the luminal surface area within the alveoli, and are predominantly responsible for gas exchange with blood vessels. Type II pneumocytes (TIIPs) secrete surfactant proteins that maintain surface tension within the alveoli. TIIPs also are a source of immunomodulatory proteins which play a role in local immune defense. In the context of osteosarcoma pulmonary metastases, alveolar cells, including TIIPs, constitute a major cell type of the lung environment[2]. There is evidence that in a mouse model, metastatic osteosarcoma cells preferentially settled into the alveolar epithelial niche[24]. In osteosarcoma patients, serum levels of pro-surfactant protein B, a protein produced by TIIPs, is elevated and significantly correlated with advanced clinical stage, metastatic disease, and reduced overall survival[25]. While not a direct representation of what is occurring within the lung microenvironment, elevated serum pro-surfactant protein B implicates a reaction from TIIPs in response to injury within the lung in the presence of lung metastases.

After injury, TIIPs proliferate to repair alveoli in a process called TIIP hyperplasia[26,27]. This is a nonspecific response to a range of lung injuries including the development of cancer[26,28]. In osteosarcoma pulmonary metastases, this injury response was found to be expressed by the alveolar epithelial cells through single cell RNA-seq, with an increase in fibrosis and other wound-healing markers surrounding the metastasis[24]. Injury or successive rounds of replication can also promote cellular senescence, inciting these cells to secrete factors that promote a pro-inflammatory and pro-tumor microenvironment[29]. Secretory factors produced by senescent TIIPs lead to fibroblast activation and development of lung fibrosis, through deposition of collagen, and enhanced osteosarcoma metastatic growth[27,30–33]. Pulmonary fibrosis increases the stiffness of the lung TME and can enhance the pro-metastatic phenotype of migrating osteosarcoma cells[34–36]. In mice with osteosarcoma, structural alterations have been demonstrated within the lung including ECM remodeling, increased deposition of fibronectin and collagen, and inflammation[36]. Therapies targeting fibrosis such as a combination of the tyrosine kinase inhibitor nintedanib and anti-fibrotic agent pirfenidone inhibit osteosarcoma metastasis to the lung in the LM8-Dunn mouse model[32] further underscoring the role of fibrosis in metastatic progression. Treatment with this combination reduced metastasis likely through inhibition of myofibroblast reprogramming, preventing the secretion of fibronectin. Collectively, this literature suggests a role for TIIPs and fibrosis in promoting a pro-tumor environment in the lung, however there is still more to be understood in the context of metastatic osteosarcoma.

In addition to promoting the development of fibrosis, lung fibroblasts can be further hijacked by the presence of metastatic osteosarcoma to produce anti-apoptotic and pro-inflammatory signals[2,19]. Among those signals are chemokines expressed by osteosarcoma cells that are implicated in the development of a pro-metastatic environment in the lung. For example, stromal expression of CXCL14 from metastatic osteosarcoma cells reprogrammed fibroblasts to a more malignant phenotype and generated a more supportive metastatic niche in the lung[33]. In an experimental mouse lung-metastasis model of intravenously injected human cell lines, the expression of growth factors by tumor-associated stromal cells within the lung pre-metastatic niche drive ERK phosphorylation and MCL1 expression[37]. Both ERK activity and MCL1 expression are found to be elevated in early mouse osteosarcoma metastases, indicating a trend toward increased proliferation and anti-apoptotic signaling in osteosarcoma tumor cells[37]. Pulmonary fibroblasts can also be directly activated by tumor-derived extracellular vesicles, inciting a transition into cancer-associated fibroblasts (CAFs) through activation of TGFβ1 and SMAD2 pathways[38]. CAFs make up a large component of the osteosarcoma lung TME and secrete a variety of factors such as chemokines, cytokines, and ECM such as collagen into the surrounding environment that create a more hospitable space for tumor cells[39]. Although the presence of osteosarcoma tumor-derived exosomes did not correlate with an increase in tumor metastases, the signaling changes in pulmonary fibroblasts and the pre-metastatic niche indicate that they play an important role in the development of pulmonary metastases[38].

Direct comparisons between the human and canine patient remain challenging, as metastatic disease in general is poorly understood. For example, not much is known about the function of TIIPs and fibroblasts in canine osteosarcoma pulmonary metastases; however, the literature showing the role of human TIIPs in metastatic osteosarcoma provides a basis for future research. In contrast, the roles of endothelial cells in metastatic progression have been investigated in both species. Vascularization and expression of the vascular endothelial growth factor (VEGF) have been implicated in promoting the development of lung metastases[40–47]. In human tumors, the presence of elevated VEGF correlates with a worse overall survival and a potential for an increase in pulmonary metastasis, and approximately 20–25% of human osteosarcomas contain a VEGFA amplification[41,42]. Of note, silencing of the protein CYR61/CCN1, a member of a family of secreted matricellular proteins, induced a reduction of pro-angiogenic markers such as VEGF[40]. Further, inhibition of CYR61/CCN1 in a mouse osteosarcoma metastasis model led to reduced tumor vasculature, decreased metastasis to the lung, and the finding that microvessel density potentially correlated with risk of metastasis[40]. The presence of microvessels surrounding primary osteosarcomas in both canine and human patients increases the risk for pulmonary metastases[43–47]. Importantly, these studies have been primarily pursued in the primary tumor context thus their role in lung metastases is less clear. In metastases in different locations, including lung, an increase in endothelial cells was correlated with decreased survival for canine osteosarcoma patients[48]. However, the TME of the primary osteosarcoma was not conserved between patient-matched metastases further emphasizing the need for a better understanding of the lung TME in both humans and dogs[48].

The development of serum biomarkers to understand disease progression and metastasis is vital for both humans and canine patients. Clinical studies of canine patients are ideally suited for the investigation of biomarkers of prognosis or therapeutic response in a longitudinal context. This is due to similarities in many facets of modern veterinary and human medical practice, which include serial blood sampling and clinical imaging studies, as well as dogs' physical size. Changes in exosomal cargo[49] and increased expression of IL-8[50] are being explored as biomarkers to indicate osteosarcoma disease progression in dogs. One study utilizing canine osteosarcoma cell lines in a mouse model identified increased expression of a p63 isoform as promoting pulmonary metastasis through interaction with VEGF-A and IL-8[51]. Of note, an increase in VEGF expression correlates with a poorer prognosis and can be used as a potential biomarker for progression[52]. In a study of 25 metastasis-free canine patients with

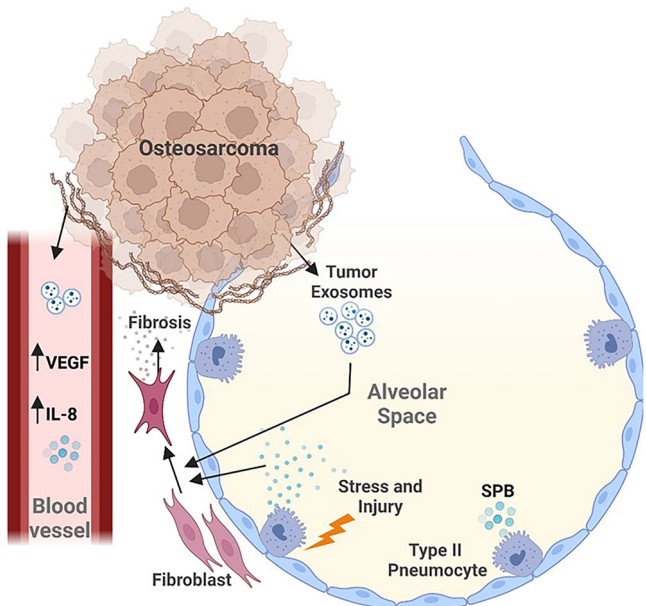

**Fig. 1 | Graphical schematic of the cellular heterogeneity present in the lung microenvironment, showing the tumor surrounding the alveolar space containing type II pneumocytes.** Osteosarcoma tumors secrete tumor-derived exosomes containing components and other factors that influence the normal activity of resident lung cells. The presence of vasculature within the tumor facilitates the dissemination of these factors and may play a role in disease progression. In response, type II pneumocytes secrete surfactant protein B (SP-B) which, in conjunction with tumor-derived exosomes, influences the development of fibrosis. Created in BioRender. Mcgee, L. (2025) https://BioRender.com/i23k019.

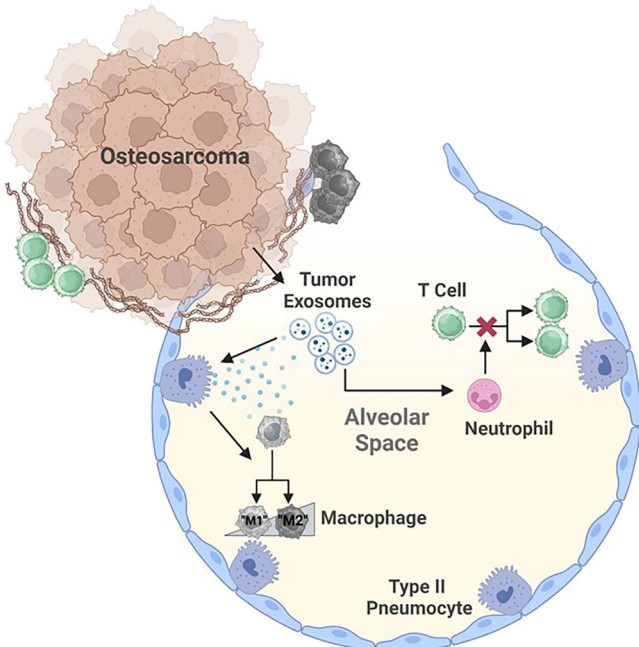

**Fig. 2 | Graphical schematic of the immune cell heterogeneity present in the lung microenvironment.** There is an accumulation of immune cells along the tumor margin, reduced T cell proliferation, prevention of antigen uptake and processing by bone marrow derived cells, and increased neutrophils within the osteosarcoma lung microenvironment. Created in BioRender. Mcgee, L. (2025) https://BioRender.com/a36o974.

osteosarcoma, elevated VEGF in pre-treated serum correlated with decreased disease free interval[52]. This may be due to increased invasive capacity and angiogenesis as has been demonstrated in primary canine osteosarcoma cell lines[53]. A study measuring primary tumor microvessel densities in canine primary osteosarcoma has shown that high vascularity may be an indication of the presence of pulmonary metastases and disease progression[43].

## Immune cells

The development of lung metastases is the single most important prognostic factor for osteosarcoma patients and is highly influenced by the interplay between the native lung microenvironment and the invading tumor cells now housed in the lung[5,18,19]. While a minority of human and canine osteosarcoma primary tumors are classified as immune-enriched[54,55], studies have shown that lymphocytes and other immune cells are often excluded from osteosarcoma metastases in the lung[56,57]. This section focuses on major immune cells within the pulmonary TME of metastatic osteosarcoma including the role of macrophages, T cells, and neutrophils, and how the canine patient can help inform the development of immunotherapies (Fig. 2).

### Macrophages

Histologically, macrophages can be observed within and around pulmonary metastases. Alveoli adjacent to metastatic lesions contain increased numbers of macrophages in both human (Fig. 3a) and canine (Fig. 3b) tumors. In human osteosarcoma tissues, macrophages infiltrate at greater density along the tumor:non-tumor interface of pulmonary metastases[56]. The functional state of these macrophages is key to understanding their role in metastatic progression. Polarization of macrophages is thought to occur as a continuum between M1 ("classically activated") and M2 ("alternatively activated")[58]. While M1 macrophages aid in the removal of tumors and activation of anti-tumor CD8+ T cells, M2 macrophages are thought to promote tumor progression through a variety of mechanisms including

promotion of angiogenesis and production of cytokines and chemokines[58–60]. Senescent pneumocytes have also been implicated in causing M2 polarization of macrophages in the lung[31]. In several osteosarcoma studies, macrophages have been shown to express classic M2 markers including IL-10 and TFGβ2[61]. For example, alveolar macrophages exposed to metastatic osteosarcoma exosomes transform into the M2 phenotype, inducing increased expression of pro-tumor IL-10 and TGFβ2[61]. A single cell RNA-sequencing study identified that *FABP4+* alveolar macrophages around osteosarcoma metastases express pro-inflammatory genes that further contribute to the immune suppressing tumor microenvironment[62]. Tumor-associated macrophages expressing CD204, a classically defined marker of M2 macrophages, have been demonstrated around osteosarcoma tumors. These tumor-associated macrophages play a role in inhibiting osteosarcoma progression and are generally associated with a better overall survival in both canine and human osteosarcoma[63,64]. The presence of macrophage precursor cells, monocytes, may also play an important role osteosarcoma metastasis[65]; in a mouse model, increased numbers of circulating monocytes prevented the development of osteosarcoma lung metastases. However, the role and importance of macrophage polarization remains poorly understood. A single cell RNA-sequencing study of pulmonary metastases identified that while macrophages were present within the samples, there was no distinct polarization detectable[2].

### T Lymphocytes

The same study that detected no distinct polarization of macrophages also described T cells as the most abundant immune cells within pulmonary metastases including CD8+ T cells expressing low levels of immune checkpoint markers[2]. In humans, pulmonary metastases harbor greater numbers of T cells surrounding the tumor compared to their matched primary tumors[56]. One study has shown the same to be true in canine OS patients[20]. Lymphocytes are frequently observed accumulating along the tumor:non-tumor interface of human (Fig. 3c) and canine (Fig. 3d) osteosarcoma metastases. T cells are capable of switching between a Th1, "helper," CD4+ T cells which aid the "cytotoxic" CD8+ T cells in

**Fig. 3 | Representative images of immune components surrounding pulmonary osteosarcoma metastases.** Human and canine tumor associated macrophages. Macrophages accumulate around the tumor margin, within alveolar spaces adjacent to **a** human and **b** canine osteosarcoma metastases. Scale Bar = 25 μm. Lymphocytes are also known to accumulate along the tumor:non-tumor interface of **c** human and **d** canine osteosarcoma metastases. Scale Bar = 100 μm.

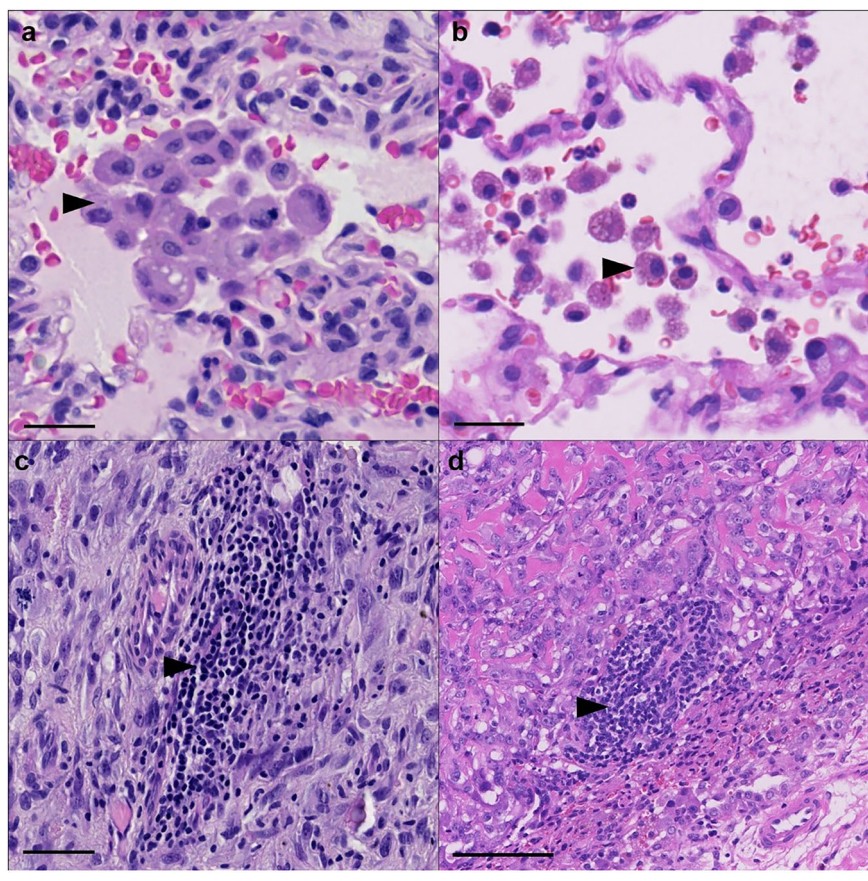

eliminating the tumor, to the Th2 phenotype which blocks the CD8+ T cells from properly eliminating the tumor[66]. In patient samples, there is an enrichment for exhausted T cell populations in the TME of OS. While particularly pronounced in the primary tumor, this immune-suppressive T cell phenotype is also observed in osteosarcoma lung metastases[62]. Inadequate T cell activation around osteosarcoma metastases is driven in part by IFNγ and can induce alternative activation in immune cells, increasing their immune-suppressive capacity[56]. Cytotoxic CD8+ T cells were further shown through tissue immunohistochemistry (IHC) to only be present directly surrounding the pulmonary metastasis, with no infiltration into the tumor detected. This study implies that initial response by tumor antigen specific T cells is halted by a variety of immune suppressive factors such as increased immune checkpoint molecules like PDL-1 and LAG-3, and suppressive myeloid cells[56]. Circulation of the PD-1 ligand, PDL-1 may also influence the progression and development of pulmonary metastases. In mice injected with the 142B osteosarcoma cell line, PDL-1 containing exosomes increased pulmonary metastasis compared to the same model in the presence of an exosomes inhibitor, GW4869[67].

**Neutrophils**

In a sequencing study of RNA collected from human osteosarcoma lung metastases, gene signatures associated with dendritic cells, neutrophils, immature myeloid cells, chemokines/cytokines, and abnormal vasculature were located around the lung metastases[56]. Bone marrow derived dendritic cells are capable of homing to future sites of metastasis and influencing the "seeding" of circulating tumor cells into the pre-metastatic niche through secretion of factors like VEGFR-1, including in the lung[68]. Neutrophils have also been reported within osteosarcoma lung metastases[56]. In addition to important anti-tumor functions, neutrophils can suppress the anti-tumor immune response and facilitate the development of the pre-metastatic niche[66,69,70]. Exosomes secreted from a metastatic mouse model increase the expression of Toll-like receptor 3 (TLR3) in lung epithelial cells (including

TIIPs), which promote a chronic inflammatory phenotype through the recruitment of neutrophils and other bone marrow-derived cells (BMDCs)[23]. The function of TLR3 and neutrophil infiltration in the pre-metastatic niche is further supported in other tumor models where TLR3 deficiency reduced metastatic burden with fewer infiltrating neutrophils[23]. Osteosarcoma cells may also contribute to this process[71]. CXCL12-containing extracellular vesicles secreted by K7M2 osteosarcoma cells induce accumulation of granulocytic myeloid-derived suppressor cells in the lung prior to the development of pulmonary metastases[71].

## Understanding immunotherapy through a comparative approach

Dogs with higher immune infiltration in their primary tumors tend to also have higher infiltration in their metastatic lesions[20]. In lung metastases, this is defined by increased infiltration by macrophages and lymphocytes as is described in human patients. Differences between human and canine immune responses must be considered to properly assess the translatability of immune therapies between human and canine patients. In a comprehensive study comparing the transcriptomic profile and function of T cells and macrophages, it was found that although general trends between species were similar, there are some key differences[15]. For instance, human T cells expressed a more defined Th1 phenotype, while the canine T cells had less distinction between Th1 and Th2; however, T cells expressed high levels of common Th1 associated genes (TBX21, RORC, GZMB, IL21) in both species[15]. In response to PHA and cytokine stimulation, canine T cells demonstrated a reduced transcriptomic response compared to human. However, canine macrophages had an increased response to cytokine secretion, suggesting a mechanism for compensation to make up for the decreased T cell response. In a separate study of canine macrophage activation, key markers for pro-inflammatory M1 and anti-inflammatory M2 macrophage activation were identified[16]. Aligned with other species, canine M1 macrophages demonstrated iNOS expression and increased bactericidal

activity, while M2 macrophages had increased expression of IL-8 and IL-10[16]. Overall, more studies are needed to fully elucidate the differences and similarities between human and canine, and how exactly the canine immune system functions, to translate findings from dogs to humans within the context of immunotherapy trials that could benefit patients with localized or metastatic disease.

Leveraging the similarities between human and canine patients, a number of immunotherapeutic approaches have been attempted with varied results[72,73]. Previous immunotherapy trials have been well described elsewhere[72,73]. More recently, trials have included the addition of interferon alfa-2b to standard of care[74] and treatment with a HER2-targeted monoclonal antibody[75]. The latter was meant to target the portion of human and canine osteosarcomas which express elevated HER2. In this clinical trial, inactivated listeria, expressing chimeric huHER2/neu, was injected into dogs after limb amputation to prime the immune system against HER2 expression. Dogs who received the vaccine experienced a median overall survival of 956 days over the historical average of 423 days[72,76]. This study led to the larger clinical trial COTC026, the results of which are pending publication. While a human osteosarcoma trial using trastuzumab to target HER2 was negative, knowledge gained from these trials can be used to inform future studies. Other canine immunotherapy trials include the use of a monocyte-recruitment blocking agent losartan to reduce metastases, with a clinical benefit found in 50% of dogs with lung metastases[77]; evaluation of an inhaled recombinant human IL-15 in dogs with osteosarcoma and melanoma lung metastases[78]; and the use of adoptive natural killer cell immunotherapy[14,79].

Another immunotherapy trial starting in canine patients used a muramyl dipeptide phosphatidylethanolamine (a synthetic analog of an immunostimulatory portion of mycobacteria cell wall) liposomal configuration (L-MTP-PE) to stimulate macrophages and monocytes[73]. Single agent dose of L-MTP-PE was able to increase median survival time[80], and serial combination of L-MTP-PE and adjuvant cisplatin treatment further increased median survival time[81]. While this treatment failed to be approved by the US FDA, it has been approved for use in the European Union as mifamurtide[73] where there has been success in increasing progression free survival in human patients with localized disease[82].

Although immunotherapy has seen recent advances across tumor types, it has not yet achieved much success in osteosarcoma[21]. A contributor to these failures could lie within the osteosarcoma microenvironment creating an environment unfavorable to immune cell infiltration[21]. With a limited human patient population pool, it is often difficult to fully understand the impact of new treatments on disease progression, but a better understanding of how the immune system interacts with osteosarcoma metastases can help improve treatments.

## Discussion

The development of metastasis to the lungs remains the primary challenge to improving survival rates for both human and canine patients[4–6]. While humans and dogs are not direct comparisons for each other, the evidence shows that canine metastatic osteosarcoma shares many of the same characteristics as human metastatic osteosarcoma. Due to the higher incidence of canine osteosarcoma, the dog is an excellent model to use to better understand and treat human osteosarcoma. With the treatment regimen for osteosarcoma remaining stagnant for the last 40 years, new treatments are desperately needed for both human and canine patients.

Some gene signatures in the dog correlate with what has been seen in the human patient population; in particular, increased VEGF expression has been correlated with poor prognosis in both patient populations[41,52,68]. These discoveries lay the groundwork for further development of shared biomarkers that could be used to inform treatment. For example, the presence and function of TIIPs in the lung responding to the presence of lung metastases is an avenue that has yet to be fully explored in both human and canine populations, yet the response of TIIPs to injury and inflammation caused by the metastases may further exacerbate the aggressiveness of the tumor[26,28]. A better understanding of the role of TIIPs in the TME and their response to treatment and disease progression is needed to further elucidate their importance.

Another similarity, the presence of immune infiltrates in the lung, informs survival in both humans and dogs[18,54,55]. In both patient populations, the presence immune infiltration correlates with a better prognosis, yet much of the TME secretes factors that suppress the functionality of the immune cells. Understanding this axis between pro- and anti-immune signals from the TME will inform the potential for new therapeutics that may promote the immune system's ability to properly target and eliminate metastases.

Further improvements in access to tumor samples, data sharing, and increased collaboration between researchers and physicians who work with human and canine osteosarcoma are required to improve outcomes for both patient populations. Since companion dogs develop osteosarcoma at a much higher incidence while sharing key similarities with human patients, they provide an optimal model with which to trial new treatments and biomarkers. As long as pulmonary metastases are the major determinant of prognosis for both patient populations, there is value in fully elucidating these similarities to advance treatments and biological understanding of the interplay between the lung microenvironment and metastatic osteosarcoma.

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

## Author contributions

L.E.M., J.S.P., and J.A.B. wrote the manuscript. T.A.M., C.M., and A.K.L. reviewed and edited the manuscript.

## Competing interests

The authors declare no competing interests.
