## [Transparent Peer Review file · Communications Biology]

The Tumor Microenvironment of Metastatic Osteosarcoma in the Human and Canine Lung

Corresponding Author: Dr Lauren McGee

Version 0:

Reviewer comments:

Reviewer #1

(Remarks to the Author)

This is a well written review summarizing the current knowledge of the tumor microenvironment, particularly as it pertains to metastases, in both human and canine osteosarcoma. There are a few minor points that the authors might consider addressing.

Line 43 – it might be useful to note that human and canine treatment protocols typically differ in the order of chemotherapy and surgery, with human patients receiving chemo first, while the vast majority of dogs receive surgery first.

Figure 3 – The rationale for this figure is unclear. The figure shows a single example of an H&E stained normal human lung, next to an example of canine lung stained for surfactant protein C. The photos are of different things, so no comparison is possible. It might be more useful to have photos of both species with both stains. In addition, the magnification of the photos varies considerably, making comparison more difficult.

Line 99-101 – This needs a reference

Line 101-103 – This appears to repeat the same information as lines 95-99

Line 115-116 – The authors should use great care in using their, or anyone else's single-cell or single-nuclei RNA seq data to quantify subgroups of cells with respect to each other. These techniques, especially single-cell RNA seq, frequently generate biased cell harvests and therefore relative quantitation of cell types is not appropriate.

Figure 4 – the magnification for human and dog are very different and this does not allow the figures to be easily compared.

There were also a number of grammatical and typographical errors:

Line 96 – numerical agreement needs to be fixed. Probably should be "...from a metastatic..."

Line 108- should be "... palmitate, a common..."

Line 151 – "altered" is redundant

Line 220 – "human" might be less awkward than "physician".

Line 222 – "likewise" is redundant.

Line 225 – should be "correlated".

Line 233 – The sentence appears to be missing a third sub-clause or should read "...in exosomal cargo and increased expression..."

Line 419 – should be "correlate"

Line 446- the incidence is not increased (that would imply that the number were growing), but rather the canine incidence is "higher" as it is being compared to humans.

Reviewer #2

(Remarks to the Author)

McGee et al. provide a review describing the comparative analyses of the microenvironment of pulmonary metastatic osteosarcoma. This manuscript describes human resident lung cells, fibroblasts and collagenous matrix, vasculature, canine resident lung cells and the ECM, human Immune cells, and canine Immune cells as they relate to metastatic OS and as an entryway into reviewing a number of molecular findings. A summary figure is provided as Figure 1. 5 immunohistochemistry figures are also included.

A review of the non-tumor derived cells present in the pulmonary metastatic OS microenvironment would be useful to the field, but the current draft, in my opinion, requires significant modification to be acceptable for publication. Much of the paper appears to be derived from findings derived from primary OS samples.

I suggest that you go through the paper and determine whether each section is relevant to the comparative aspects of OS pulmonary metastases. The following suggestions are not complete and are provided to improve the paper.

1) The structure of the review needs to be tightened so that each aspect of the metastatic environment and what is known from humans and canines need to be addressed in a consistent fashion. The current structure is difficult to follow and seems quite piecemeal.

2) Many sections diverge from describing the tumor microenvironment of metastatic osteosarcoma and focus on patient outcome associations in primary tumors. Many claims of outcome associations exist in the literature for primary tumors, with varying levels of replication. Describing a subset of these is not very helpful to the central idea of the review.

3) Figure 2 (immunohistochemistry image) does not provide a meaningful addition to the paper in the current form. Specifically, the following statement does not need an image of 4 dog metastatic tumors to support. The rationale for each of the remaining figures should be carefully considered.

“The lung is the most common site of metastatic progression in people and in dogs with osteosarcoma (Figure 2)”

4) The writing needs to be carefully edited. Citations exist where findings and conjecture are liberally mixed. (i.e lines 31-33) “In human patients, survival rates have plateaued over the last 40 years, due mainly to inadequate treatment of patients with metastatic disease, which occurs in up to 80% of patients 1-3, 7

5) Run-on sentences are present which mix separate statements for which the comparative aspects are not further discussed. These two points should be separated and the presence of micro-mets in humans should be discussed.

“It is likely that 80-90% of dogs present with micro-metastases in the lung at diagnosis, contributing to a 3-year survival rate that hovers around 10%, however it should be noted that this endpoint is skewed as many owners choose euthanasia when the disease progresses after treatment 6, 8, 9.”

6) The last 3 sentences of the first paragraph of the discussion should be in the introduction.

7) The final paragraph of the discussion should focus on metastatic osteosarcoma.

8) If you describe comparative immunotherapy in OS I suggest reviewing

Wycislo, K. L., and T. M. Fan. “The immunotherapy of canine osteosarcoma: a historical and systematic review.” *Journal of veterinary internal medicine* 29.3 (2015): 759-769. Along these lines Mepact should be mentioned if you discuss immunotherapeutic approaches.

9) A discussion of single cell OS RNA-Seq as it pertains to OS metastasis should also include the plethora of human studies, not only the canine data in isolation.

10) In the vasculature section a summary statement that “controversy exists” should be inserted before the description of specific positive and negative associations between microvessel density and outcomes that contradict each other.

11) In line 222 a prior discussion of VEGF is described which does not exist.

“As previously discussed in human patients, an increase in VEGF expression likewise correlates with a poorer prognosis in canine patients and can be used as a potential biomarker for progression.”

Version 1:

Reviewer comments:

Reviewer #1

(Remarks to the Author)

The authors have been very responsive to the original reviews. They have made substantial changes to the manuscript which focus and streamline it, making it acceptable for publication, overall. The revised material does contain a number of grammatical and typographical errors that should be addressed prior to publication. Some examples are:

Line 202- 203 “role” and “have” disagree with respect to singular vs. plural

Line 218 – perhaps “...metastases to different locations...” substituting to for in.

Line 413 – “...number of in both...” delete the “in”

Line 426 – “...identified THAT FABP4+...” Insert “that”

Line 446 – “...true in the canine OS patients...” delete “the”

Line 459-460 – it should probably be PD-L1, not PD-1

Line 543-546 – this should be made into two sentences.

29 January 2025

Dear Reviewers,

Thank you to you for taking the time to review our article. Below you will find the point by point responses to the concerns that were raised. After extensive restructuring, we hope the content of this review is clearer. Specifically, we have restructured the sections of the review so that they flow more smoothly. The sub-sections under “Resident Lung Cells” have been collapsed so they are less disparate and are therefore easier to follow. We have also removed literature primarily focused on primary osteosarcoma tumors, as well as other tumor types that had relevant biology to osteosarcoma pulmonary metastases but were not directly related.

Further, we have removed figures 2, 3, and 4 from the review. The figure 1 infographic has been split into two figures to highlight the non-immune and immune components in the two halves of the review. Figures 5 and 6 have been combined into 1 figure to demonstrate the accumulation of immune cells surrounding osteosarcoma pulmonary metastases in both human and canine samples.

We welcome your feedback and hope you are satisfied with our revisions.

Warm regards,

Lauren McGee
Jessica Beck
Amy LeBlanc

National Cancer Institute, NIH
10 Center Drive
Bethesda, MD 20892-4258

Regarding the figures which were identified as a concern by both reviewers, please:

- **update Fig. 3 and 4 with comparable images**
- place all original **microscopy images (Fig. 2, 3, 4, 5, 6) within a single figure or supplementary figure** and reference them appropriately in the main text so that their relevance is reflected
- **simplify Fig. 1** and emphasize the specific processes depicted in the circular inset; consider removing the coloured background to make the text, arrows and small pictures of cells more clearly visible, and also explain the figure briefly in the main text.

We strongly recommend **adding more summarising infographics** which would reflect the content of the review and make it easier to follow and understand.

We are removing figures 2, 3, and 4 and collapsing Figures 5 and 6 into one figure. We are also editing Figure 1 and splitting it into two figures, showcasing “non-immune” and “immune” components in the lung TME. New figures are included at the end of this document.

Reviewer #1 (Remarks to the Author):

This is a well written review summarizing the current knowledge of the tumor microenvironment, particularly as it pertains to metastases, in both human and canine osteosarcoma. There a few minor points that the authors might consider addressing.

Line 43 – it might be useful to note that human and canine treatment protocols typically differ in the order of chemotherapy and surgery, with human patients receiving chemo first, while the vast majority of dogs receive surgery first.

A statement to this effect has been added (lines 44-46: “However, while human patients receive chemotherapy prior to amputation, canine patients undergo amputation before starting any chemotherapeutic treatments.”)

Figure 3 – The rationale for this figure is unclear. The figure shows a single example of an H&E stained normal human lung, next to an example of canine lung stained for surfactant protein C. The photos are of different things, so no comparison is possible. It might be more useful to have photos of both species with both stains. In addition, the magnification of the photos varies considerably, making comparison more difficult.

This figure has been removed

Line 99-101 – This needs a reference

Reference to Liu, Y, *et al.* has been added to this statement, and moved to the “Neutrophils” section of the review. (Lines 487-489)

Line 101-103 – This appears to repeat the same information as lines 95-99

Second mention of the mouse model has been removed

Line 115-116 – The authors should use great care in using their, or anyone else’s single-cell or single-nuclei RNA seq data to quantify subgroups of cells with respect to each other. These techniques, especially single-cell RNA seq, frequently generate biased cell harvests and therefore relative quantitation of cell types is not appropriate.

Removed references to exact cell populations (lines 131-133)

Figure 4 – the magnification for human and dog are very different and this does not allow the figures to be easily compared.

This figure has been removed

There were also a number of grammatical and typographical errors:

Line 96 – numerical agreement needs to be fixed. Probably should be “...from a metastatic...”

Fixed

Line 108- should be “... palmitate, a common...” Fixed

Line 151 – “altered” is redundant Fixed

Line 220 – “human” might be less awkward than “physician”. Fixed

Line 222 – “likewise” is redundant. Fixed

Line 225 – should be “correlated”. Fixed

Line 233 – The sentence appears to be missing a third sub-clause or should read “...in exosomal cargo and increased expression...” Fixed

Line 419 – should be “correlate” Fixed

Line 446- the incidence is not increased (that would imply that the number were growing), but rather the canine incidence is “higher” as it is being compared to humans. Fixed

Reviewer #2 (Remarks to the Author):

McGee et al. provide a review describing the comparative analyses of the microenvironment of pulmonary metastatic osteosarcoma. This manuscript describes human resident lung cells, fibroblasts and collagenous matrix, vasculature, canine resident lung cells and the ECM, human Immune cells, and canine Immune cells as they relate to metastatic OS and as an entryway into reviewing a number of molecular findings. A summary figure is provided as Figure 1. 5 immunohistochemistry figures are also included.

A review of the non-tumor derived cells present in the pulmonary metastatic OS microenvironment would be useful to the field, but the current draft, in my opinion, requires significant modification to be acceptable for publication. Much of the paper appears to be derived from findings derived from primary OS samples.

I suggest that you go through the paper and determine whether each section is relevant to the comparative aspects of OS pulmonary metastases. The following suggestions are not complete and are provided to improve the paper.

1) The structure of the review needs to be tightened so that each aspect of the metastatic environment and what is known from humans and canines need to be addressed in a consistent fashion. The current structure is difficult to follow and seems quite piecemeal.

We appreciate the feedback and have worked to adjust the layout to be more cohesive. After re-working the draft, we hope you find it that it now has better structure and flow.

2) Many sections diverge from describing the tumor microenvironment of metastatic osteosarcoma and focus on patient outcome associations in primary tumors. Many claims of outcome associations exist in the literature for primary tumors, with varying levels of replication. Describing a subset of these is not very helpful to the central idea of the review.

Extraneous references to the primary tumor that do not have direct implications to the pulmonary metastases have been removed. (Ex: lines 108-112, 120-129, 259-275, 336-349)

3) Figure 2 (immunohistochemistry image) does not provide a meaningful addition to the paper in the current form. Specifically, the following statement does not need an image of 4 dog metastatic tumors to support. The rationale for each of the remaining figures should be carefully considered.

“The lung is the most common site of metastatic progression in people and in dogs with osteosarcoma (Figure 2)”

This figure has been removed

4) The writing needs to be carefully edited. Citations exist where findings and conjecture are liberally mixed. (i.e lines 31-33)

“In human patients, survival rates have plateaued over the last 40 years, due mainly to inadequate treatment of patients with metastatic disease, which occurs in up to 80% of patients 1-3, 7

These issues have been corrected (line 38)

5) Run-on sentences are present which mix separate statements for which the comparative aspects are not further discussed. These two points should be separated and the presence of micro-mets in humans should be discussed.

“It is likely that 80-90% of dogs present with micro-metastases in the lung at diagnosis, contributing to a 3-year survival rate that hovers around 10%, however it should be noted that this endpoint is skewed as many owners choose euthanasia when the disease progresses after treatment 6, 8, 9.”

This issue has been corrected (lines 46-48)

6) The last 3 sentences of the first paragraph of the discussion should be in the introduction.

These sentences have been added to the introduction (lines 76-82)

7) The final paragraph of the discussion should focus on metastatic osteosarcoma.

We have added better wording to emphasize metastatic osteosarcoma (lines 649-656)

8) If you describe comparative immunotherapy in OS I suggest reviewing Wycislo, K. L., and T. M. Fan. "The immunotherapy of canine osteosarcoma: a historical and systematic review." *Journal of veterinary internal medicine* 29.3 (2015): 759-769. Along these lines Mepact should be mentioned if you discuss immunotherapeutic approaches.

Included a reference to previous reviews of comparative immunotherapies as well as a section on mifamurtide/Mepact (lines 560-567). In addition, we included the canine immunotherapy review from Wycislo, *et al.* as reference 73.

9) A discussion of single cell OS RNA-Seq as it pertains to OS metastasis should also include the plethora of human studies, not only the canine data in isolation.

References to single cell sequencing of human pulmonary metastases have been made clearer (ex: references 24, 62)

10) In the vasculature section a summary statement that “controversy exists” should be inserted before the description of specific positive and negative associations between microvessel density and outcomes that contradict each other.

This section has been re-worked to more directly focus on pulmonary metastases and less so on prognostic significance. (lines 200-222)

**11) In line 222 a prior discussion of VEGF is described which does not exist.
“As previously discussed in human patients, an increase in VEGF expression likewise correlates with a poorer prognosis in canine patients and can be used as a potential biomarker for progression.”**

This has been corrected

Figure 1. Graphical schematic of the cellular heterogeneity present in the lung microenvironment, showing the tumor surrounding the alveolar space containing type II pneumocytes. Osteosarcoma tumors secrete tumor-derived exosomes containing components and other factors that influence the normal activity of resident lung cells. The presence of vasculature within the tumor facilitates the dissemination of these factors and may play a role in disease progression. In response, type II pneumocytes secrete surfactant protein B (SP-B) which, in conjunction with tumor-derived exosomes, influences the development of fibrosis. Figure was developed using Biorender.

Figure 2. Graphical schematic of the immune cell heterogeneity present in the lung microenvironment. There is an accumulation of immune cells along the tumor margin, reduced T cell proliferation, prevention of antigen uptake and processing by bone marrow derived cells, and increased neutrophils within the osteosarcoma lung microenvironment. Figure was developed using Biorender.

Figure 3. Representative images of immune components surrounding pulmonary osteosarcoma metastases. Human and canine tumor associated macrophages. Macrophages accumulate around the tumor margin, within alveolar spaces adjacent to **a)** human and **b)** canine osteosarcoma metastases. Scale Bar = 25 μm . Lymphocytes are also known to accumulate along the tumor:non-tumor interface of **c)** human and **d)** canine osteosarcoma metastases. Scale Bar = 100 μm .

REVIEWERS' COMMENTS:

Reviewer #1 (Remarks to the Author):

The authors have been very responsive to the original reviews. They have made substantial changes to the manuscript which focus and streamline it, making it acceptable for publication, overall. The revised material does contain a number of grammatical and typographical errors that should be addressed prior to publication. Some examples are:

Line 202- 203 “role” and “have” disagree with respect to singular vs. plural

Fixed, now on line 151

Line 218 – perhaps “...metastases to different locations...” substituting to for in.

Fixed, now on line 163

Line 413 – “...number of in both...” delete the “in”

Fixed, now on line 199

Line 426 – “...identified THAT FABP4+...” Insert “that”

Fixed, now on line 212

Line 446 – “...true in the canine OS patients...” delete “the”

Fixed, now on line 230

Line 459-460 – it should probably be PD-L1, not PD-1

Fixed, now on line 243

Line 543-546 – this should be made into two sentences.

Fixed, now on line 293-296